# Explosive percolation yields highly-conductive polymer nanocomposites

Manuela Meloni [1], Matthew J. Large [1] ✉, José Miguel González Domínguez [2], Sandra Victor-Román[2], Giuseppe Fratta[1], Emin Istif[2], Oliver Tomes[1], Jonathan P. Salvage[3], Christopher P. Ewels [4], Mario Pelaez-Fernandez[5], Raul Arenal [5,6,7], Ana Benito [2], Wolfgang K. Maser [2], Alice A. K. King [1], Pulickel M. Ajayan[8], Sean P. Ogilvie [1] ✉ & Alan B. Dalton [1] ✉

Explosive percolation is an experimentally-elusive phenomenon where network connectivity coincides with onset of an additional modification of the system; materials with correlated localisation of percolating particles and emergent conductive paths can realise sharp transitions and high conductivities characteristic of the explosively-grown network. Nanocomposites present a structurally- and chemically-varied playground to realise explosive percolation in practically-applicable systems but this is yet to be exploited by design. Herein, we demonstrate composites of graphene oxide and synthetic polymer latex which form segregated networks, leading to low percolation threshold and localisation of conductive pathways. In situ reduction of the graphene oxide at temperatures of <150 °C drives chemical modification of the polymer matrix to produce species with phenolic groups, which are known crosslinking agents. This leads to conductivities exceeding those of dense-packed networks of reduced graphene oxide, illustrating the potential of explosive percolation by design to realise low-loading composites with dramatically-enhanced electrical transport properties.

The segregated network approach is a technically-simple way of using hierarchical assembly to produce functional micro- and nanostructures with low filler loadings, by comparison to isotropic random mixtures. Segregated networks may be formed using colloidal dispersions of filler and matrix particles[1], such as polymer latex and nanosheets of graphene or other nanomaterials (with applications in coatings), or other colloidal particles such as those of silicon, with carbon nanotubes forming the filler in state-of-the-art battery electrodes[2]. The excluded volume created by the matrix particles during drying confines the filler particles to the interstitial space,

where the excluded volume can be modified by tailoring the particle size distribution[3]. This method is an increasingly common way to produce nanocomposites[4] as it gives control of the structure at the nanoscale.

The networks of conductive particles formed during the preparation of segregated networks often display properties consistent with percolation theory. Percolation theory describes the non-linear scaling of transport properties with loading level in statistically-filled systems due to the random nature of particle-particle connections. A universal scaling relation for electrical conductivity $\sigma$ according to

[1]University of Sussex, Brighton BN1 9RH, UK. [2]Instituto de Carboquímica (ICB-CSIC), Zaragoza 50018, Spain. [3]School of Pharmacy and Biomolecular Science, University of Brighton, Brighton BN2 4GJ, UK. [4]Institut des Materiaux Nantes Jean Rouxel, Nantes, France. [5]Laboratorio de Microscopias Avanzadas (LMA), Instituto de Nanociencia de Aragon (INA), U. Zaragoza, Mariano Esquillor s/n, 50018 Zaragoza, Spain. [6]Instituto de Ciencias de Materiales de Aragon, CSIC-U. de Zaragoza, Calle Pedro Cerbuna 12, 50009 Zaragoza, Spain. [7]ARAID Foundation, 50018 Zaragoza, Spain. [8]Department of Materials Science and Nanoengineering, Rice University, Houston, TX, USA. ✉e-mail: m.large@sussex.ac.uk; s.ogilvie@sussex.ac.uk; a.b.dalton@sussex.ac.uk

percolation theory relates a parameter describing the conductivity of the filler network $\sigma_0$, the loading level $\phi$, the percolation threshold loading $\phi_c$ and scaling exponent $\mu$ by[5]:

$$\sigma = \sigma_0 (\phi - \phi_c)^{\mu}$$

It is well understood that systems wherein the filler is distributed randomly have a scaling exponent which is dictated only by the dimensionality of the domain upon which the particles are distributed $-\mu \approx 1.33$ in 2D systems, such as thin coatings, and $\mu = 2$ in 3D systems, such as bulk composites. Many experimental reports of nanocomposites fall within these two limits[6]. Deviations from the universal values are associated with changes to the distribution of conductances within the system[5], as well as changes to the spatial distribution of the particles. In the former case, broadening of the inter-particle contact resistance distribution (for example) leads to a monotonic increase in $\mu$ above the universal value;[5] exponents more than double the expected value[7], and even as high as 11.9[8], have been reported for experimental systems. In the latter case, where the spatial distribution of particles in the system displays strong local correlations, a reduction in $\mu$ below the universal value may be obtained. In this case, the addition of new particles leads to aggregation with existing particles, leading to the growth of dense conductive clusters. Once the percolation threshold is reached, a dense system-spanning conductive network is formed with high conductivity, that increases only marginally with the presence of additional particles (corresponding to a low value of $0 < \mu < 1$). This has been described as explosive percolation in theoretical studies[9], and is characterised by a relatively high percolation threshold (delayed conductivity onset) and a low critical exponent, when compared to isotropic percolative systems[9].

Latices have been increasingly well studied for the formation of composites that are variously transparent, flexible, and conductive using metal powder[10,11] or carbon black[12] as fillers. In addition to manipulation of the nanostructure, latex-based systems have the added benefit that no volatile organic components (VOCs) are released during film formation. Indeed, such nanocomposite systems have been used as platforms for investigating the predictions of percolation theory in an experimental space. The influence of latex size distribution, and aspect ratio and size distributions of the nanomaterial fillers[5] on properties such as the percolation threshold have been examined by previous work, including that of Grunlan and co-workers[13] in 2001 with carbon black, and more recently by Jurewicz et al. using carbon nanotubes[14] and graphene nanosheets[15,16].

Graphene oxide (GO), a chemically-modified graphene derivative, can be readily exfoliated to produce laterally-large monolayer sheets with a high aspect ratio. GO has the additional benefits of commercial availability at a large scale, low cost, and may be processed entirely in water, making it highly compatible with a latex-based composite approach. However, GO requires thermal and/or chemical reduction to realise appreciable electrical conductivity[17]. It is well-documented in the literature for individual flakes, as well as films and membranes, that temperatures greater than 300 °C, and often in excess of 700 °C, are usually required to achieve a sufficiently high degree of reduction for conductive percolating networks of rGO to form[18-21]. The studies cited, as well as many others, demonstrate that reduction at lower temperatures causes only partial removal of oxygen functionalities, accompanied by the formation of topological defects which require annealing for a high-quality sp² structure to be obtained.

We note, however, that the reduction of GO is a partially-exothermic process. This is since the decomposition of epoxy basal plane functionalities, which are metastable, releases significant heat. As such, there is potential for the reduction of GO to be used as a platform for in situ chemistry, even at the relatively low temperature of ~200 °C which causes decomposition of epoxy functionalities[22,23].

In this work, we demonstrate a GO-latex composite system with anomalously-high electrical conductivity, processed using a mild (<150 °C) thermal treatment to initiate reduction of the GO. Through microscopic, macroscopic, and spectroscopic methods, we elucidate aspects of the mechanism by which a seemingly dense, low-loading conductive network evolves during a partial GO reduction process. We anticipate that this methodology has the potential to be a general route toward high-conductivity, low-loading networks for a range of applications.

## Results and discussion
### Segregated network nanocomposites
Figure 1A, B show schematics of the formation of a latex-GO composite system (full experimental methods are given in the Supplementary Information). The film formation approach is similar to that in refs. 15, 24. The latex is a copolymer of methyl methacrylate (MMA), butyl acrylate (BA) and methacrylic acid (MAA); used for its relatively low glass transition temperature ($T_g = 20$ °C) and stability at temperatures up to 200 °C. Figure 1C shows a representative AFM of a dried composite film, where the individual polymer spheres remain visible. The samples are then annealed at up to 150 °C under vacuum to achieve mild reduction of the GO in situ, after which a macroscopic colour change from brown to black is observed, with an onset of electrical conductivity at GO loadings as low as 0.5 wt%. After this process, the polymer particles fully coalesce, as seen in the AFM of Fig. 1D.

Figure 1E plots the conductivity of latex-GO composites as a function of GO content, after the brown-black colour change is observed when the samples are heated. The inset photographs further illustrate the change in the appearance of the composites through several of the processing steps. Unlike conventional percolative networks, a very sharp increase from the baseline conductivity of the matrix polymer to a saturating network conductivity is observed.

The observations of both a rapid increase above the percolation threshold $\phi_c$, combined with a saturation of conductivity $\sigma$, suggest that the percolation exponent of the system, $\mu$, is low (0.6), which is indicative of an explosive percolation mechanism. To illustrate the comparison of the present system with an isotropic percolative system of particles with a comparable aspect ratio (in the same matrix polymer), Supplementary Information Fig. 1 plots normalised percolation curves for the data in Fig. 1E along with data from our prior work[15].

We note that the saturation conductivity of the composites is greater than that achieved for a bulk rGO film processed in the same manner (see Fig. 1E). From this observation, we can infer that mobility changes must occur within the composite network, since the free-carrier density is inherently linked to the rGO content of the system and therefore would linearly increase with GO content.

Based on the concept of explosive percolation, we would expect to observe dense networks of conductive material within the bulk of the sample, even at loadings just above the percolation threshold. Figure 1F shows a contrast-enhanced cross-sectional SEM of a fractured composite at 2 wt% GO loading. Clearly visible is a filamentous network of conductive pathways which appears to be uniform at scales greater than several hundreds of nanometres. This agrees with the notion that the conductive networks may have 'grown' from the originally-seeded segregated network structure formed by the GO nanosheets within the polymer matrix.

The visible structure of the network, as well as the length scale of the observed pathways, is inconsistent with the structure of the originally-included GO nanosheets (see Supplementary Information). This implies the formation of conductive material from, or through interaction with, the polymer matrix during the reduction process. Investigation of the bulk mechanics (Fig. 1G) of the composites shows no statistically-significant change before and after annealing, for either the pristine polymer or the 2 wt% GO composite, yielding highly conductive ($R < 10\,\Omega$) yet mechanically-robust polymeric materials. We

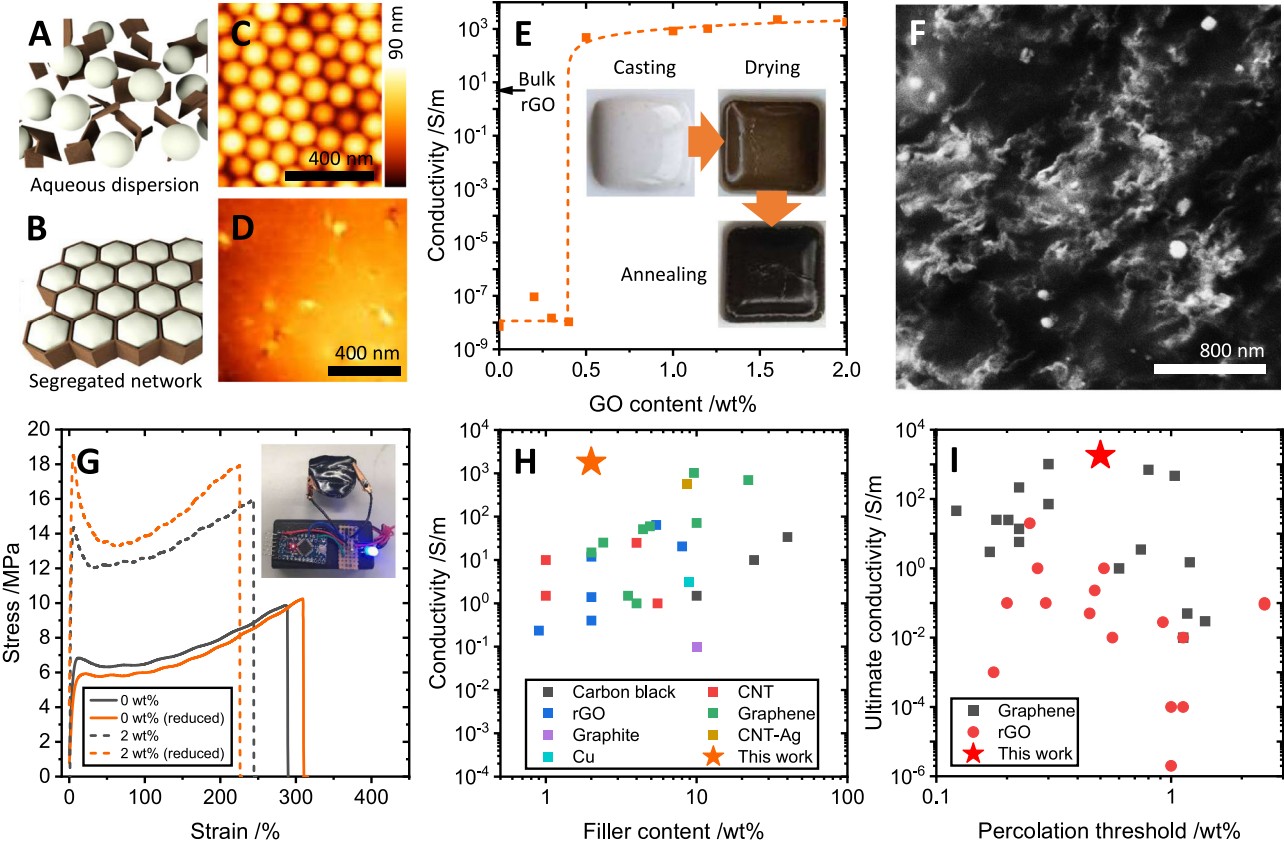

**Fig. 1 | Assembly and properties of segregated network nanocomposites.**
**A**, **B** Schematics of the composite latex-GO dispersion and subsequently-templated film structure. **C**, **D** Atomic force micrographs of a latex film before and after heating above the glass transition temperature of the polymer, resulting in particle coalescence. **E** Achieved conductivity in the latex-GO composites post-reduction. Indicated is the conductivity of a bulk rGO film processed under the same conditions. (inset) Photographs of the sample at each processing step. **F** Scanning electron micrograph of a fracture surface of a post-reduction composite.

**G** Representative stress-strain curves for pristine latex films and 2 wt% latex-GO composites, before and after reduction, showing toughening under the addition of GO to the polymer, but no significant change in mechanics post-reduction.
**H** Comparison of the maximum reported conductivity (and corresponding filler content) for 25 composites from the literature, categorised by filler material. A table of references is available in the Supplementary Information. **I** Maximum reported conductivity of polymer composites of rGO and pristine graphene from the literature against reported percolation threshold.

can infer from this that chemical modification of the polymer is strongly localised, and that bulk carbonisation is not responsible for the changes in conductivity observed. If local carbonisation surrounding the reducing GO sheets in situ were occurring, then the interfacial interactions responsible for the increased toughness, Young's modulus would be modified, which is not observed in the data.

Figure 1H illustrates a comparison of the conductivity at 2 wt% achieved in this work, to the maximum conductivities and loading levels achieved in 28 polymer-nanoparticle composite systems from the literature, classified by filler type. We see that the present latex-GO composites, after reduction, are substantially more conductive than other systems at comparable loading levels, and even with systems at an order of magnitude higher filler loading. Figure 1I illustrates that this high conductivity is not exclusively a property of the segregated network structure. In general, a more strongly segregated network will present a lower percolation threshold, which increases the conductivity at a fixed loading above that threshold. In Fig. 1I, we see that the percolation threshold in this work falls within the range of many others reported in the literature for both rGO and pristine graphene composites. However, the conductivity achieved is substantially higher than the ultimate conductivities achieved in all rGO composite systems, and is competitive with pristine graphene systems with comparable percolation thresholds (albeit at higher loading levels, as seen in Fig. 1H). This deviation from the broad trend of rGO composite behaviour suggests that the conductive network present in our system

is not exclusively defined by tunnelling conduction as is the case in other "physical" rGO and pristine graphene networks.

## In situ reduction of graphene oxide
Based on the data in Fig. 1, we anticipate that the GO reduction is the driving influence behind changes to the system. The reduction of GO is associated with a colour change from brown to black, as shown in the inset to Fig. 1E. The time series data in Fig. 2A–C show that, for samples with a GO content above the percolation threshold identified in Fig. 1E, the colour change is also associated with a sharp onset of electrical conductivity. This behaviour is particularly indicative of the sharp conductivity transition associated with growing networks in explosive percolation problems[9]. Changes to the reduction temperature also reveal an inverse modification to the time at which the conductivity onset occurs.

Figure 2D plots the time at which the colour change and conductivity onset are observed against the inverse absolute temperature of reduction. We observe an Arrhenius-type behaviour, with two exponential regions indicative of energy-activated processes with two distinct activation energies. The functional form of the two fitted curves is $\ln[\text{time}] = \ln\left[A\exp(E_A/k_B T)\right]$, where $E_A$ is the process activation energy, $k_B$ is the Boltzmann constant and $T$ is the absolute temperature. The two processes have activation energies of $E_A = 0.36$ eV or ~35 kJ/mol (from 40 to 135 °C) and $E_A = 1.45$ eV or 140 kJ/mol (from 150 to 220 °C where the temperature is sufficient for GO to undergo exothermic reduction[22]). For the latter process, the

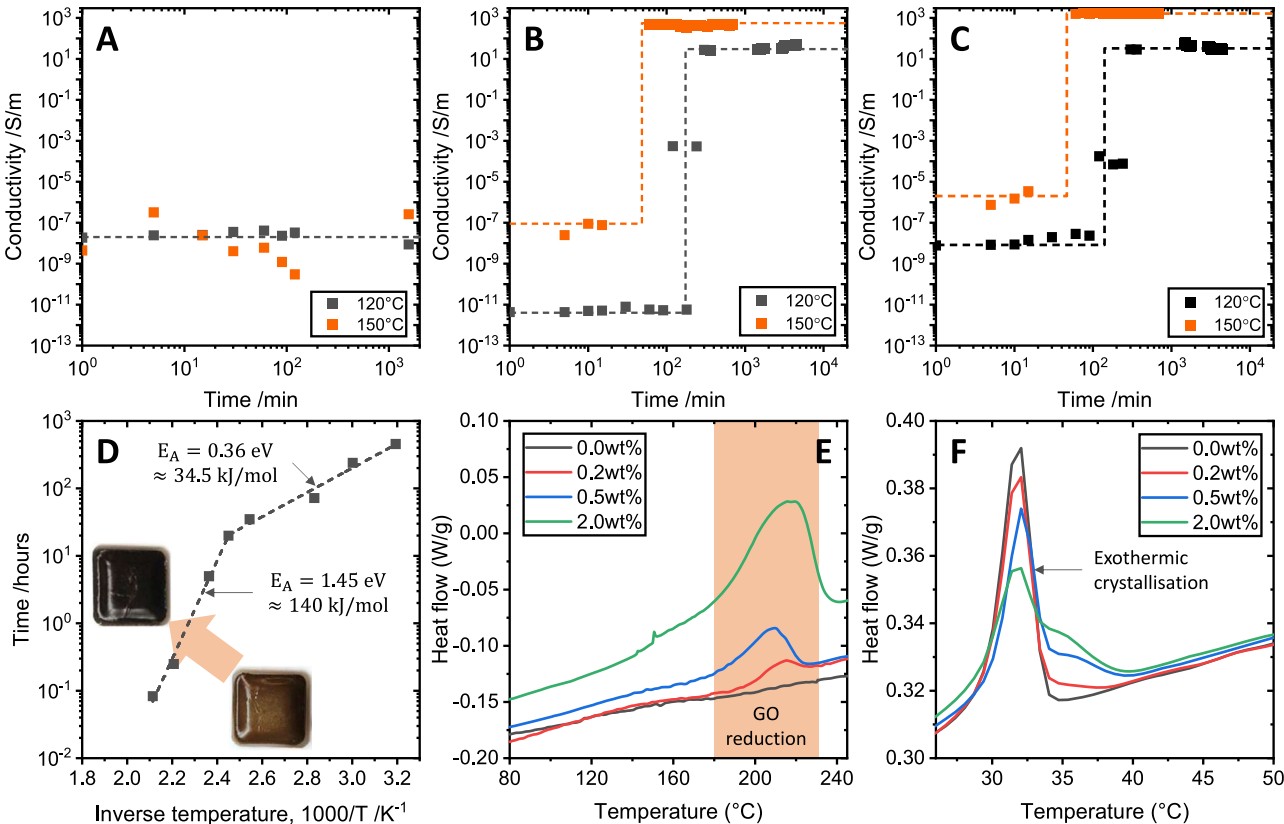

**Fig. 2 | Time- and temperature-dependent properties.** Conductivity time series for a below-percolation-threshold composite (**A** 0.2 wt% GO), a near-threshold composite (**B** 0.5 wt%) and above-threshold composite (**C** 2 wt%) reduced at two different temperatures. **D** Arrhenius plot of the time taken for both a colour change and conductivity onset to occur, as a function of inverse absolute temperature, showing the presence of two distinct regions with their associated activation energies. **E** DSC traces for the pristine polymer and three composites (0.2, 0.5, and 2.0 wt% GO) show the exothermic GO reduction peak at around 200 °C. **F** DSC traces show the evolution of the exothermic polymer crystallisation peak with increasing GO content.

activation energy coincides with that reported for bulk GO thermal decomposition[23]. Differential scanning calorimetry (DSC) results in Fig. 2E confirm that there is an exothermic process that appears as a peak between 150 and 220 °C; this feature appears only in the composites that contain GO, and matches that observed for GO reduction reported elsewhere[22].

Also observed in the DSC data are changes to the exothermic polymer crystallisation peak, shown in Fig. 2F (the corresponding DSC range showing endothermic melting is presented in the Supplementary Information). For the composites, after reduction, the feature associated with the pristine polymer (at 32 °C) reduces in intensity and a second feature at a higher temperature (at 35 °C) emerges, with prominence correlated with the rGO content of the composite. We suggest that this is a result of modified polymer crystallinity emerging at the interface with the GO sheets within the composite[25].

Figure 3 shows the results of microscopic analyses aimed at investigating any morphological and electronic changes corresponding to the observed macroscopic behaviours detailed in Figs. 1, 2. Spin-coated thin-film samples were characterised before and after thermal treatment using Kelvin probe force microscopy (KPFM), SEM and X-ray photoelectron spectroscopy (XPS). Figure 3A, B show KPFM topography and contact potential difference (CPD) data for an unreduced composite sample. The structure of the polymer particles is visible prior to heating, however, the CPD data (which is the difference in work function between the sample surface and KPFM probe) shows no structure. After reduction, the polymer particles become fully coalesced (as evidenced in the topography data in Fig. 3C) and a substantial structure emerges in the CPD map (Fig. 3D).

Figure 3E plots histograms of the pixel-wise work function, $\Phi_{sample} = \Phi_{tip} - CPD$, for the data in Fig. 3B, D. As can be observed, initially, the work function distribution is very narrow, and at a value close to that of the Pt-Ir-coated tip (5.1 eV). This is consistent with GO which is reported to have a work function >5 eV[26]. Post-reduction, the distribution broadens to lower work function values, consistent with the calculated ionisation potentials of small-molecule conjugated carbon systems in the literature, as well as for rGO whose work function falls within the range 5 to 4.4 eV (depending on the extent of reduction)[26].

The structural features which correspond to the lower work functions in the post-reduction sample in Fig. 3C, D are at length scales of ~100 nm, which appears compatible with the conductive features observed in the SEM of Fig. 1F. Figure 3F shows SEM at a directly-comparable resolution to the KPFM data, illustrating that the electron-dense structures visible in the imaging are compatible with the electronic features observed. This combination of techniques suggests that the structures observed are likely to be part of the conducting network in the sample. Further, the length scales are too small, and the features too uniformly distributed, to be rGO (see Supplementary Information for details of the GO nanosheets used). We conclude that these networks are a consequence of the in situ GO reduction process, which leads to the high conductivities observed at the macroscale.

## Crosslinking of percolating networks

In order to ascertain the precise nature of the nanoscopic networks observed, x-ray photoelectron spectroscopy (XPS) and Fourier transform infra-red (FTIR) spectroscopy were performed on the samples, both pre- and post-reduction. The results are collated in Fig. 4, where

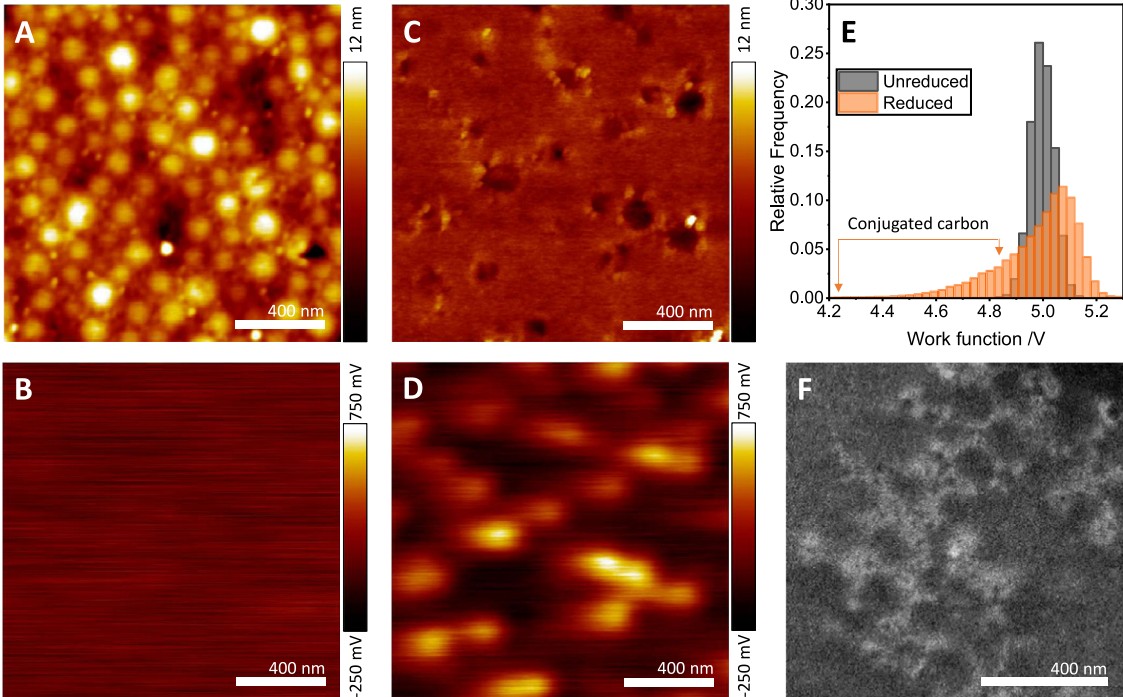

**Fig. 3 | Nanoscale topographic and electronic characterisation.** Kelvin probe force microscopy (KPFM) data for the topography (**A**) and contact potential difference (**B**) of an unreduced composite; panels **C**, **D** show the corresponding measurements after reduction of the GO in situ. **E** Modification of the work function of the system calculated from the CPD maps in **B**, **D**. **F** SEM of a fracture cross-section showing a dense conducting network, with features on the same length scale as the pre-reduction polymer latex particles.

the fitted XPS spectra for the C1$s$ and O1$s$ peaks are shown in Fig. 4A–D, E–H, respectively, and the FTIR spectra before and after reduction are shown in Fig. 4I.

Deconvolution of the XPS data is performed using four components in the C1$s$ peak and three in the O1$s$ peaks in the unreduced systems, based on analysis of the polymer molecular structure. Components were assigned to CH$_n$ groups (i.e. $sp^3$ carbon atoms, 285.0 eV), beta-shifted C-COO groups (carbons adjacent to a carboxylic or ester group, 285.7 eV), C-O groups (alcohol, ether, epoxy groups, 286.8 eV), and COO groups (carboxylic acid, ester, 289.1 eV) in the C1$s$ spectra. Components fitted to the O1$s$ spectra were assigned to O=C-O and O=C groups (carbonyl oxygens in carboxylic acid and ester groups, 532.1 eV), COC and C-O-H groups (epoxy and alcohol oxygens, 532.8 eV), and O=C-O-C groups (ester oxygen, 533.8 eV). We note that the anticipated feature due to $sp^2$ content of the GO is obscured by the large $sp^3$ content of the polymer and the noise in the measurements, given the GO content of the sample is only 2 wt%.

Upon heating under the conditions used in this study, the pristine polymer becomes almost completely reduced, as observed in the C1$s$ spectra (Fig. 4A, B). The O1$s$ spectra for the same samples imply the reduction of all carbonyl oxygens to alcohols, with the possible survival of etheric oxygens. This process appears to result in the formation of an almost entirely aliphatic polymer system.

When GO is added to the system to form a composite, the C1$s$ (Fig. 4C, D) and O1$s$ (Fig. 4G, H) spectra show the emergence of new components. In the unreduced C1$s$ spectrum (Fig. 4C), we observe the presence of the same components as in the pristine polymer, although with different peak ratios. In the heated sample, there is a stronger retention of carboxylic/ester and alcohol/ether groups and a new feature emerges at 284.0 eV; this new peak is consistent with $sp^2$ carbons found in graphitic materials[27], indicating the emergence of a conjugated carbon component at levels substantially larger than expected based on the initial GO inclusion. Indeed, the $sp^2$ component

one would expect based on the presence of GO is broadly undetectable in the unheated samples.

Before heating, the O1$s$ spectra for the GO composite is comparable to that for the pristine sample; as observed in the C1$s$ comparison for the sample samples, there is a slight variation in relative peak areas. Post-heating all detectable carbonyl oxygens are lost from the composite sample, as is the case with the pristine sample (Fig. 4E, F). However, a second feature emerges at 531.7 eV, which is not observed in the heated pristine polymer sample. This feature may be attributed to carbonyl and hydroxyl groups in esters, lactones and anhydrides[28,29], however, it is also consistent with phenolic hydroxyl groups[30,31], which can be rationalised in tandem with the C1$s$ data showing the emergence of $sp^2$ carbon in the system, and with the KPFM of Fig. 3E which shows the presence of material with work functions consistent with small conjugated molecules.

The FTIR measurements in Fig. 4I corroborate the XPS analysis; after heating, small features emerge at ~1580 cm⁻¹ (consistent with $sp^2$ carbon systems) and in the range 700 to 1050 cm⁻¹, which are consistent with aromatic C-H modes (out-of-plane bending modes near 700 cm⁻¹ and in-plane bending modes near 1000 cm⁻¹) such as those observed in benzene[32]. While the macroscopic temperature applied to the samples is relatively low, local temperatures close to the reducing GO sheets may be higher. It is understood that heating PAA and PMAA to 400 °C causes polymer pyrolysis with aromatics as the dominant species[33].

In the context of the high conductivity of the reduced samples, the presence of possible conjugated polyol molecules is important. Phenolic crosslinking of GO has been demonstrated in various studies[34–37]. Additionally, the use of di-functional conjugated molecules has been shown to substantially increase the conductivity of films and aerogels of various layered materials. In the work of Worsley et al.[38] benzenediol was used to crosslink a low-density (1 vol.%) graphene oxide aerogel, leading to a factor of 200 increase in the electrical conductivity of the samples. Similarly, ref. 39. show an order of

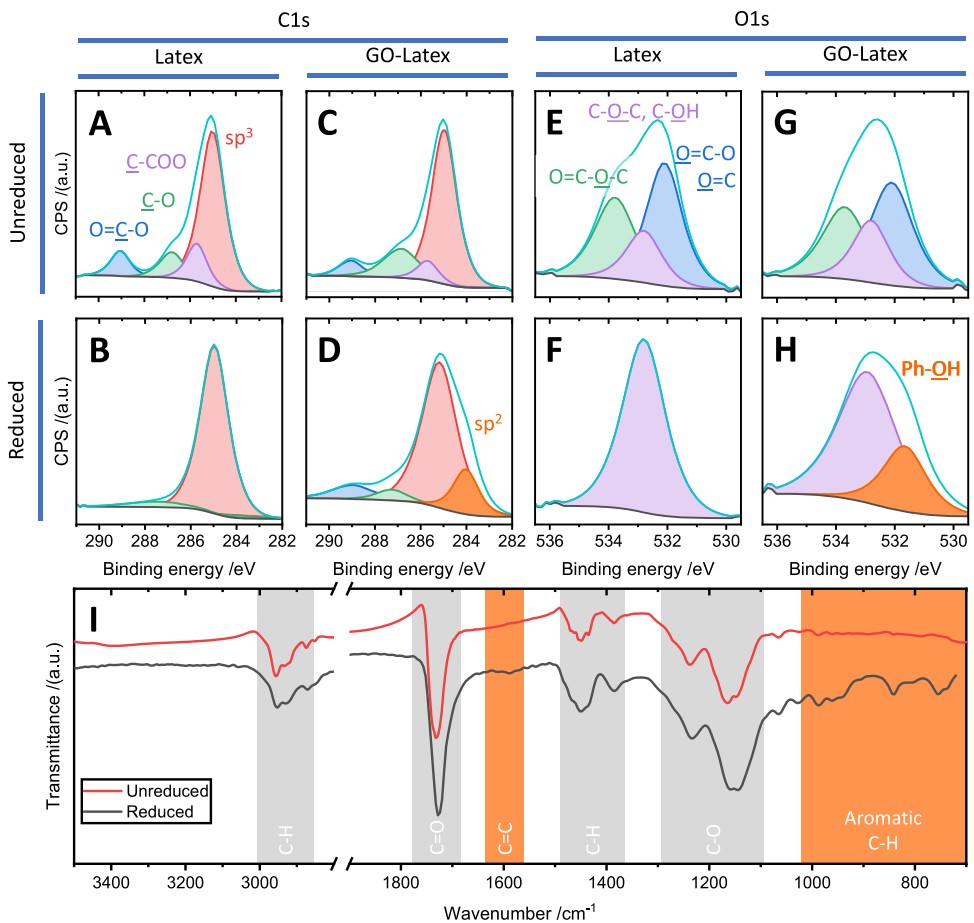

**Fig. 4 | Chemical spectroscopy of nanocomposites. A–H** Fitted XPS data showing the C1s (**A–D**) and O1s (**E–H**) features. The unreduced (**A, C, E, G**) and reduced (**B, D, F, H**) samples are investigated for the pristine latex polymer and 2 wt% GO composites. **I** FTIR spectra for both unreduced and reduced 2 wt% composites. Mode assignments are indicated, with those emerging as a result of the reduction process highlighted in orange.

magnitude increase in the conductivity of nanosheet networks of molybdenum disulfide when crosslinked in the presence of benzene dithiol, which was demonstrated to attach at defect sites on the nanosheets.

Accurate determination of a precise reaction mechanism underlying the conductive network formation is challenging in the system studied due to its chemical richness, and uncertainties surrounding the decomposition of the matrix copolymer and how intermediates and products may interact with the reducing graphene oxide and one another. In broad terms, however, we propose that the process involves the decomposition of AA/PAA to OH-terminated conjugated molecules, as have been demonstrated to covalently crosslink GO/rGO in other studies. These conductive small molecules form additional electron pathways in the system which 'grow' from the reducing GO (which itself acts as a strongly-localised heat source for the thermal decomposition), forming densely-connected macroscopic networks when the initial GO inclusion is sufficiently high (i.e. at and above the percolation threshold).

In summary, we report a polymer-nanomaterial composite system with electrical properties at low loading levels that outperforms both its own bulk filler and most other polymer-nanomaterial composites at comparable loading. A study of both the macroscopic thermal, chemical and electrical properties, along with microscopic structural investigation, reveals the formation of a system-scale conductive network consisting of both reduced graphene oxide filler particles as well as conjugated small molecules formed during the in situ reduction process.

The attractive characteristics of this material system are realised by the combined influence of the three separate factors. Firstly, the segregated network approach confers a low percolation threshold and co-localises the filler particles to the interstitial sites. Secondly, in situ exothermic reduction of the graphene oxide results in the thermal decomposition of the adjacent polymer molecules. This, in turn, produces phenolic species which act as crosslinking agents for the confined reduced graphene oxide network to finally realise high conductivity at low filler loadings while retaining the flexibility of the polymer matrix.

Building on this understanding, we anticipate that it is readily possible to formulate filler systems comprising nanomaterials and corresponding crosslinking precursors. Such an approach would facilitate an extension of the presented chemistry beyond latex coatings, with transferability into other templated systems such as composites of ceramics, metals and thermosetting polymers. The results presented here have important implications for the production of functional coatings, such as for electromechanical strain sensors, as well as for multifunctional structural composites.

This work illustrates the potential of a by-design explosive percolation approach to preserve the macroscopic performance of templated matrix materials while harnessing the functional properties of embedded covalent nanomaterial networks.

## Methods
### Polymer latex
The latex polymer used was provided by DSM NeoResins (Waalwijk, The Netherlands) and is a copolymer of butyl acrylate (BA), methyl

methacrylate (MMA), and methacrylic acid (MAA). The polymer particle size is 172 nm, its dry glass transition temperature (Tg) is 20 °C, and the initial solids content is 54 wt%. The latex dispersion was prepared by semi-batch emulsion polymerisation using sodium dodecyl sulfate (SDS) to achieve stability and ammonium persulfate as the initiating salt per descriptions elsewhere[40,41].

### Graphite oxide preparation
Graphite oxide was prepared using a modified Hummers method as reported elsewhere[20,42]. In detail, 5 g of graphite flakes were put into a mixture of 170 mL $H_2SO_4$ and 3.75 g $NaNO_3$ and the mixture was cooled in an ice bath and stirred for 30 min. Thereafter, 25 g of $KMnO_4$ were slowly added and stirred for another 30 min. Then, the ice bath was removed, and the mixture was warmed up to 35–40 °C and stirred for 2 h. The reaction was terminated by adding slowly 250 mL of distillate water and then 20 mL $H_2O_2$ (30%) solution. The mixture was stirred for 2, 4 h or overnight depending on the oxidation degree, filtered and the obtained powders were repeatedly washed with 400 mL of $HCl:H_2O$ (1:10 v/v) to remove metal ions, followed by distillate water to remove the acid. In the end, the graphite oxide obtained was dried at room temperature.

### Graphene oxide (GO) preparation
Graphene oxide was obtained by bath sonication of an aqueous graphite oxide dispersion (2 mg/mL) for 2 h, followed by centrifugation at 4500 rpm (2332×$g$) for 60 min, obtaining a brown-coloured dispersion of exfoliated GO with a final concentration of 1 mg/mL. The GO solution was dispersed in DI water to achieve 0.5 mg/mL of GO concentration.

### Composite preparation
Graphite Oxide made by modified Hummer's method were dispersed in deionized water and sonicated using a water bath for 1 h to produce a dispersion of Graphene Oxide. A bimodal, polymer latex sample prepared by emulsion polymerisation was used. Depending on the desired GO concentration in the final composite mixture (ranging from 0.2 to 2 wt%), latex dispersion was combined with aqueous GO dispersion. All the samples were then sonicated in the water bath for 10 min to ensure good mixing. Calculation of the amount of GO in weight percent was based on the assumption that GO is in solution and on the basis of the weight of the latex solids content.

### Electrical characterization
The measurements were performed in two ways: (a) using a multimeter and (b) using a two-point probe system (Keithley 2614B) and the IV characteristics were measured through the gold electrodes or gold wires for the free-standing sample. The final specific conductivity was calculated from the resistance and thickness of the film. The polymer films were prepared by casting the composite dispersion onto the gold electrodes or casting in the free-standing mould adding gold wires. In both cases, the sample was then dried for 24 h at room temperature.

### AFM
For topological studies, an Atomic Force Microscope Dimension® icon Bruker positioned in an insulated box over an anti-vibrant stage to minimise environmental noise and building vibrations was employed. Bruker Scan Assist-Air AFM probes Silicon Tip on the Nitride lever have an average spring constant of 0.4 N/m and a tip radius of 2 nm (resonance frequency of 70 kHz). Scan Assist-Air AFM probes were used to acquire surface topographic images. The latex dispersion and the composites dispersions were drop casted onto a glass slide and depending on the type of latex investigated, the solution was left to dry for 24 h at room temperature. The dry samples were then placed on the AFM stage. PeakForce KPFM measurement were performed with a Dimension Icon system, using a PFQNE-AL tip with spring constant calibrated with standard contact calibration on silica at around 0.6 N/m. These tips are pure silicon super sharp tips with a tip radius of around 5 nm and proprietary reflecting coating on the backside. The contact potential difference (CPD) measurement was obtained during the interleave passage following the previously measured line topography at a set distance of 80 nm in air. The latex dispersion was drop casted onto a glass slide and it was left drying for 24 h at room temperature. Subsequently, the GO dispersion was spin-coated onto the latex substrate, followed by the reduction step using 150 °C for 5 h. The sample were then placed on the AFM stage for KPFM measurement.

### DSC
Measurements were made in a Mettler DSC-823e equipment, calibrated using an indium standard (heat flow calibration) and an indium-zinc standard (temperature calibration). The experiments were performed on samples of about 4–5 mg, exactly weighed, placed into standard 40 μL Aluminium crucibles, under a 100 mL/min flow of $N_2$. A first heating isotherm (50 °C for 5 min and cooling down to room temperature at −10 °C/min) was applied in order to erase the thermal history of the polymer matrix. Then, a heating programme was performed from room temperature to 250 °C (at 10 °C/min) and back down to room temperature (at 10 °C/min). This programme was executed up to three consecutive cycles.

### SEM
Microstructural investigations of composite materials were made using a Zeiss SIGMA field emission gun scanning electron microscope (FEG-SEM) at an accelerating voltage of 1–1.5 kV in a working distance of 2.1 mm. Samples were imaged in high vacuum conditions and a secondary electron detector was used for image acquisition. The pre-reduction samples could not be successfully imaged due to their highly-insulating nature, with significant beam damage and charging noted during measurements. Owing to the relatively high electrical conductivity of the post-reduction samples, they were imaged successfully without modification.

### XPS analysis
X-ray photoelectron spectroscopy (XPS) measurements were performed in a SPECS Sage HR 100 spectrometer with a non-monochromatic Mg X-ray source with a Kα line of 1253.6 eV energy and 300 W. An electron flood gun was used in order to neutralise the surface charging. The fitting of the XPS data was carried out by using the CasaXPS software. In all cases, Shirley-type backgrounds were applied, and all deconvoluted components were Gaussian/Lorentzian mixed shape (with 70% Lorentzian character), except for C=C $sp^2$ components, which were treated with an asymmetry function typical of graphitic structures[43].

### FTIR analysis
The FTIR measurements were performed using PerkinElmer Spotlight 400 FTIR Microscope System. The system uses dual mode single point and MCT array (mercury cadmium telluride) detector standard with InGaAs array option for optimised NIR imaging. All the measurements were done using the mid-IR (4000–500 cm$^{-1}$). For the free-standing specimens, all the measurements have been done using the ATR mode in reflectance (from 3500–700 cm$^{-1}$).

## Data availability
The processed data used in this study are available in the Sussex Figshare database under https://doi.org/10.25377/sussex.21388851. Unprocessed data were available upon request from the corresponding author.

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

## Acknowledgements

This work was funded by the European Union's Horizon 2020 research and innovation programme under the Marie Skłodowska-Curie grant

agreement no. 642742. A.B.D. acknowledges EPSRC Capital Award EP/S018069/1. W.K.M. and A.B. acknowledge support from Spanish MCIN/AEI under project PID2019-104272RB-C51/AEI/10.13039/501100011033 and the Government of Aragon (DGA) under project "Grupos Consolidados" T03-20R.

## Author contributions

M.M., M.J.L., S.V.-R., O.T. and A.A.K.K. prepared the composites and performed electrical and mechanical measurements. J.M.G.D., E.I., A.B. and W.K.M. performed and analysed calorimetry and x-ray photoelectron spectroscopy. G.F., J.P.S., M.P.-F. and R.A. performed and analysed the microscopy. C.P.E. and P.M.A. analysed the chemical spectroscopy. M.J.L., S.P.O. and A.B.D. led the analysis and interpretation and prepared the manuscript.

## Competing interests

The authors declare no competing interests.
