## [Peer Review File · Nature Communications]

Explosive percolation yields highly-conductive polymer nanocompositesReviewers' Comments:

Reviewer #1:

Remarks to the Author:

In this manuscript, authors reported a highly conductive nanocomposite with a low nano-filler content which can be prepared by a facile method. However, the "Explosive percolation" mentioned in this paper has been studied and reported in many articles, which in another word, segregation structure, just as cited by the author in the article. The characterization of structure and performance in this paper is not innovative and systematic as well.

1. The abstract describes the "Explosive percolation" as the highlight, but the mechanism and related research of this theory are not mentioned in the introduction, which causes poor reading experience. In addition, the abstract also thinks that the reduction of graphene oxide in this paper adopts mild temperature. However, a reaction temperature of 200°C can hardly be considered mild.

2. It is mentioned in the paper that Fig. 1F shows a clear conductive network, but in fact it is difficult to distinguish the conductive network from the matrix material in the figure.

3. The mechanism of the reaction between GO and the matrix material was studied in the last part of the paper. However, clear reaction mechanism and results are not provided.

Therefore, I am negative to the publication of this manuscript in Nature Communications.

Reviewer #2:

Remarks to the Author:

The authors have generated a graphene oxide (GO) segregated network within an acrylic emulsion with very low percolation threshold and very high electrical conductivity. Achieving > 10 S/cm at a loading below 1 wt% rGO is indeed impressive. AFM, SEM, DSC, and KPFM are all used correctly to support assertions/conclusions made regarding the source of this high conductivity and low loading. The text is well-written and the methodology is sound throughout. This is an interesting study that sheds some new light on segregated network nanocomposites and "explosive percolation". With that said there are some key questions that should be answered/addressed before this manuscript is suitable for publication:

1. On the third page of the manuscript, in the second paragraph, the authors say "Unlike conventional percolative networks...". This is an overstatement because all percolative networks exhibit a very sharp jump in electrical conductivity at the percolation threshold, especially in segregated networks. This is the definition of percolation as it relates to electrically conductive composites.

2. In the last paragraph on page 5, the authors mention "polymer crystallinity", but it is doubtful that a random acrylic copolymer crystallizes at all. There is no corresponding melting peaks in the DSC curves shown in Figure 2. I don't know what the observed DSC peaks are from, but it's not crystallization of the acrylic copolymer. This needs to be addressed/modified in the text. If there is crystallization of the polymer, why is there no melting observed at higher temperature (i.e. inverse peak in the DSC should be observed 20 - 40 degrees higher). Was this observed and simply not shown? It should at least be shown in Supplemental Information.

3. The authors refer to Supplemental Information as "Supplementary Information". This should be corrected.

Reviewer #3:

Remarks to the Author:

This manuscript reports on an a fairly simple process to produce a conductive polymer composite with very low conductive filler. The observation of such a high conductivity at the 0.5% loading is

remarkable for a polymer/GO composite. The authors also do a good job to investigate the phenomenon with a wide range of characterization techniques and with well-reasoned, clear analysis and discussion. The conclusions are well supported by the data. This work should be of interest to a broad audience. Therefore I recommend acceptance for publication after minor revisions listed below.

1. Please discuss a bit more on why there is no sp² peak visible in the XPS spectrum of the unheated GO/polymer composite. This is very curious and deserves more discussion.
2. Please add more details about the latex sphere preparation to ensure that the work can be reproduced by other researchers.

Reviewer 1

1. *...the "Explosive percolation" mentioned in this paper has been studied and reported in many articles, which in another word, segregation structure, just as cited by the author in the article.*
2. *The abstract describes the "Explosive percolation" as the highlight, but the mechanism and related research of this theory are not mentioned in the introduction, which causes poor reading experience.*

We apologise for the confusion that has arisen due to the unclear structuring of our introduction. Explosive percolation and percolation in segregated composites are identified as subtly different phenomena in this work. A system displaying explosive percolation characteristics may or may not be prepared in a segregated composite, and a segregated composite does not necessarily display the characteristics of explosive percolation.

Explosive percolation is characterised by a reduction in the percolation scaling exponent, and an increase in the percolation threshold filler content, of a system upon moving from an isotropic, uncorrelated spatial arrangement of filler particles, to a spatial distribution that displays strong local correlations (i.e. aggregation of the filler). In the former case, at the percolation threshold a sparse network with few conducting pathways is formed. The conductivity increases according to a power law with a well-established universal scaling exponent. In explosive percolation, dense conducting clusters of particles are formed below the percolation threshold which, when connected at the percolation threshold, produce a densely-connected, high-conductivity macroscopic network. This occurs at a higher filler loading, but increases in filler content do not yield as rapid an increase in conductivity as is observed in the "isotropic" case (i.e. the percolation exponent is markedly reduced).

We have modified and expanded our discussion of this distinction in the main text, and brought it forward to the introductory section of the manuscript. Additionally, a comparison of the conductivity data for the present system with an analogous system which exhibits isotropic percolation has been added to the Supporting Information to add further clarity.

3. *In addition, the abstract also thinks that the reduction of graphene oxide in this paper adopts mild temperature. However, a reaction temperature of 200 °C can hardly be considered mild."*

We note that the temperature ranges studied for the reduction process in this manuscript, generally below 200 °C, can be considered "mild" or "gentle" in the context of more common thermal reduction approaches for graphene oxide materials. It is well-documented in the literature for individual flakes, as well as films and membranes, that temperatures greater than 300 °C, and often in excess of 700 °C, are required to achieve a sufficiently high degree of reduction for conductive percolating networks of rGO to form [1-4]. The studies cited, as well as many others, demonstrate that reduction at lower temperatures causes only partial removal of oxygen functionalities, accompanied by formation of topological defects which require annealing for a continuous, conductive sp² structure to be obtained.

We believe that our use of the phrase "mild" is justified in the context of the literature cited, and have added additional citations to the manuscript to reflect this. We have also

made clear the temperature range of reduction in the abstract to remove ambiguity in the phrasing of the text.

[1] Detailed thermal reduction analyses of graphene oxide via in-situ TEM/EELS studies, M. Pelaez-Fernandez et al., *Carbon* 2021, 178, 477-487 (DOI: 10.1016/j.carbon.2021.03.018)

[2] In-situ reduction by Joule heating and measurement of electrical conductivity of graphene oxide in a transmission electron microscope, S. Hettler et al., *2D Materials* 2021, 8, 031001 (DOI: 10.1088/2053-1583/abcdc9)

[3] Flexible conductive graphene paper obtained by direct and gentle annealing of graphene oxide paper, C. Valles et al., *Carbon* 2012, 50, 835-844 (DOI: 10.1016/j.carbon.2011.09.042)

[4] Graphene oxide-carbon nanotube hybrid assemblies: cooperatively strengthened OH...O=C hydrogen bonds and the removal of chemisorbed water, J.D. Nunez et al. *Chemical Science* 2017, 8, 4987-4995 (DOI: 10.1039/C7SC00223H)

Reviewer 2

1. *On the third page of the manuscript, in the second paragraph, the authors say "Unlike conventional percolative networks...". This is an overstatement because all percolative networks exhibit a very sharp jump in electrical conductivity at the percolation threshold, especially in segregated networks. This is the definition of percolation as it relates to electrically conductive composites.*

We would refer back to the discussion above associated with the related comments from Reviewer 1 on explosive percolation. We believe that the modifications and additions made remove the ambiguity of the statement on the sharpness of the percolative transition region.

2. *It is mentioned in the paper that Fig. 1F shows a clear conductive network, but in fact it is difficult to distinguish the conductive network from the matrix material in the figure.*

The conductive network appears as a bright filamentous structure in the SEM image, with the matrix polymer forming a relatively uniform background. In order to improve clarity, we have adjusted the image contrast.

3. *The mechanism of the reaction between GO and the matrix material was studied in the last part of the paper. However, clear reaction mechanism and results are not provided.*

We note that accurate determination of a precise reaction mechanism is challenging in the system studied due to its chemical richness, and uncertainties surrounding the decomposition of the matrix copolymer and how intermediates and products may interact with the reducing graphene oxide and one-another. As such, we have aimed to elucidate key points of the mechanism which are informed by, and help explain, the phenomenology of the system and the evolution of the macroscopic physical properties of the composites.

In broad terms, we believe that the process involves decomposition of AA/PAA to OH-terminated conjugated molecules, which have been demonstrated to covalently cross-link GO/rGO in other studies. These conductive small molecules form additional electron pathways in the system which "grow" from the reducing GO (which itself acts as a strongly-localised heat source for the thermal decomposition), forming densely-connected macroscopic networks when the initial GO inclusion is sufficiently high (i.e. at and above the percolation threshold).

We have incorporated the above clarification into the discussion of our results in the main text.

4. *In the last paragraph on page 5, the authors mention "polymer crystallinity", but it is doubtful that a random acrylic copolymer crystallizes at all. There is no corresponding melting peaks in the DSC curves shown in Figure 2. I don't know what the observed DSC peaks are from, but it's not crystallization of the acrylic copolymer. This needs to be addressed/modified in the text. If there is crystallization of the polymer, why is there no melting observed at higher temperature (i.e. inverse peak in the DSC should be observed 20-40 degrees higher). Was this observed and simply not shown? It should at least be shown in Supplemental Information.*

The polymer is semi-crystalline, since extended DSC data does show a melting transition on the heating cycle at a higher temperature than the peak we have attributed to

crystallisation. Data have been added to the SI to support this statement, with appropriate discussion in the main text. We further note that the presence of GO/rGO causing a shift in the crystallisation temperature of crystalline polymers is well-reported in the literature [5], and have added a citation to the main text to this effect.

[5] Reduced Graphene Oxide-Induced Polyethylene Crystallization in Solution and Nanocomposites, S. Cheng et al., *Macromolecules* 2012, 45, 2, 993–1000 (DOI: 10.1021/ma2021453)

Reviewer 3

1. *Please discuss a bit more on why there is no sp² peak visible in the XPS spectrum of the unheated GO/polymer composite. This is very curious and deserves more discussion.*

We suggest that the GO sp² peak is obscured due to the very large sp³ contribution to the XPS from the matrix polymer; the relative inclusion of GO in the sample is 0.2 to 2wt%, and so we anticipate that the sp² peak is within the measurement noise of the sample. We have noted this in the main text to improve clarity of the XPS discussion.

2. *Please add more details about the latex sphere preparation to ensure that the work can be reproduced by other researchers.*

The latex polymer used was provided by DSM NeoResins (Waalwijk, The Netherlands) and is a copolymer of butyl acrylate (BA), methyl methacrylate (MMA), and methacrylic acid (MAA). The polymer particle size is 172 nm, its dry glass transition temperature (T_g) is 20°C, and the initial solids content is 54wt %. The latex dispersion was prepared by semi-batch emulsion polymerization using sodium dodecyl sulfate (SDS) to achieve stability and ammonium persulfate as the initiating salt per descriptions elsewhere [6-7].

These details have been added to the Supporting Information.

[6] Importance of Capillary Forces in the Assembly of Carbon Nanotubes in a Polymer Colloid Lattice, I. Jurewicz et al., *Langmuir* 2012, 28, 21, 8266-8274 (DOI: 10.1021/la301296u)

[7] Film Formation of Latex Blends with Bimodal Particle Size Distributions: Consideration of Particle Deformability and Continuity of the Dispersed Phase, A. Tzitzinou et al., *Macromolecules* 2000, 33, 7, 2695-2708 (DOI: 10.1021/ma991372z)

Reviewers' Comments:

Reviewer #1:

Remarks to the Author:

I still have several question.

1. Why is the method in this work superior to other similar methods? The authors should provide convincing explanation and evidence.
2. The reason for the successful thermal reduction of GO at low temperature in this work is not clarified.
3. Figure 1F: SEM image of the section of the material before thermal reduction is missing. In fact, the comparison of SEM images before and after thermal reduction can more intuitively show that the "Explosive percolation" is constructed in the material during the thermal reduction process.
4. The introduction should be modified to improve the logical coherence and readability of the article.

Reviewer #2:

Remarks to the Author:

The authors appear to have adequately addressed all reviewer comments.

Reviewer #3:

Remarks to the Author:

Authors has adequately responded to reviewer comments. I recommend publication

1. Why is the method in this work superior to other similar methods? The authors should provide convincing explanation and evidence.

We thank the reviewer for requesting this important clarification. We believe that the importance of the methodology presented here is two-fold: we demonstrate observations consistent with a mathematical phenomenon (explosive percolation) which is not readily observed in materials systems, and we find that the outcome is a composite system which significantly out-performs other polymer nanocomposites at comparable loadings *by virtue* of the explosive percolation phenomenon and related chemical modification of the matrix polymer in the system (Figure 1H of the main text highlights this comparison).

We have demonstrated a polymer nanocomposite which exhibits low percolation threshold and sharp percolation transition to conductivity exceeding that of the filler itself. These attractive characteristics are realised by the combined influence of the three separate factors in the design of the system. Firstly, the segregated network approach confers low percolation threshold and co-localises the filler particles to the interstitial sites. Secondly, *in situ* exothermic reduction of the graphene oxide results in thermal decomposition of the adjacent polymer spheres. This, in turn, produces phenolic species which act as crosslinking agents for the confined reduced graphene oxide network to finally realise high conductivity at low filler loadings while retaining the flexibility of the polymer matrix. This has important implications for production of robust functional composite materials, for example, for electromechanical strain sensors, as well as for multifunctional structural materials such as glass-fibre- and carbon-fibre-reinforced composites.

We have highlighted this analysis and interpretation of the attractiveness and applicability of the approach and material system in the conclusions of the revised manuscript.

2. The reason for the successful thermal reduction of GO at low temperature in this work is not clarified.

We note that similar points were raised in previous reviews surrounding the nature of the low temperature reduction of graphene oxide in this work. Graphene oxide is well known to undergo mild thermal reduction, specifically decomposition of metastable basal plane epoxy groups at the relatively low temperatures used here, as per references 18-21 in the main text. Our work utilises this partial, and importantly exothermic, reduction mechanism to drive further decomposition and modification of the matrix polymer. The emergent species lead to crosslinking of the partially-reduced graphene oxide, which results in a significant conductivity enhancement irrespective of the degree of the reduction of the constituent graphene oxide sheets.

3. Figure 1F: SEM image of the section of the material before thermal reduction is missing. In fact, the comparison of SEM images before and after thermal reduction can more intuitively show that the "Explosive percolation" is constructed in the material during the thermal reduction process.

Before reduction, the polymer composites exhibit very low macroscopic conductivity and therefore are challenging to image by SEM without coating. However, sputter coating with metal would mask the underlying structure of the composites which is the feature we wish to investigate. We have performed SEM on unreduced composite samples but observe significant charging and beam damage, preventing satisfactory imaging of the samples before reduction, as shown below (left scale

bar 100 μm , right scale bar 200 μm . We have noted the challenges in imaging these samples in the revised Supporting Information and highlighted the AFM of the structure of the pre-reduction samples in Figures 1 and 3 of the main text.

4. The introduction should be modified to improve the logical coherence and readability of the article.

We have revised the structure of the introduction such that the narrative comprises: introduction of segregated networks, percolation and explosive percolation therein, and their assembly from latex and nanocarbons. This leads into identification of graphene oxide as a chemically-rich filler material, challenges around the need for its reduction for conductivity but potential of this exothermic process to initiate *in situ* chemical modification and its relation to this work. We believe these changes have significantly improved the clarity and this reflects the structure of the subsequent discussion of the outcomes of this study.